# Research on the dynamic changes of China's agro-processing industry agglomeration and spatial impact of production factors on agglomeration

Hui Lu[1,2], Qing Li[2], Bin Liu[1]*, Zexin Chi[3]*, Yongmei Ye[4]

1 Research Center of "Agriculture, Rural Areas and Farmers", Jiangxi Agricultural University, Nanchang, China, 2 Institute of Agricultural Economics and Information, Jiangxi Academy of Agricultural Sciences, Nanchang, China, 3 Jiangxi Academy of Agricultural Sciences, Nanchang, China, 4 School of Economics and Management, Jiangxi Agricultural University, Nanchang, China

* liubin@jxau.edu.cn (BL); Clintonczx@163.com (ZC)

**Data Availability Statement:** All relevant data are within the paper and its Supporting Information files.

## Abstract

Agglomeration is an important phenomenon that accompanies with a large concentration of various production factors. Since the agro-processing industry has become a vital bridge connecting the primary and secondary industries, agglomeration and distribution within this industry are arousing wide concern. Based on provincial panel data of China from 2001 to 2020, this study described the dynamic changes of China's agro-processing industry agglomeration using spatial Gini coefficient and average concentration ratio. A theoretical analysis framework was established and a dynamic spatial Durbin model was used to quantitatively explore the spatial effects of production factors on the agro-processing industry agglomeration, results showed that: first, the agglomerations of agro-processing industry and its sub-industries all have exhibited a fluctuating trend of "up-down-up-down", meanwhile the agro-processing industry shifted from the Eastern coastal region to the central and western regions. Second, in the short term, the capital, labor, and technology factors respectively have remarkably promoted the agro-processing industries agglomeration both in local and neighboring areas. Third, in the long term, these three production factors all had a crowding effect on industrial agglomeration. Therefore, it is necessary to continue optimizing the agro-processing industry spatial layout through adjusting production factor inputs to promote high-quality development of the agro-processing industry.

## Introduction

With the prosperity from China's opening up to the outside world, the southeastern coastal areas, by virtue of their advantageous geographical location, have continuously attracted the inflow of capital, labor, technology production factors and exploited their great "inducing effect", forming a conspicuous phenomenon of industrial agglomeration since the 1980s [1]. However, since 2004, with the excessive concentration of manufacturing industries, resource

**Funding:** This research was funded by National Natural Science Foundation of China, grant number 17LZUJBWZX010 (Z.X. C.); System of Vegetable Industrial Technology of Jiangxi Province, grant number XARS-06 (Q. L.); Project for Key Research Ttems of Humanities and Social Science for University in Jiangxi Province, grant number JD21080 (B. L.); Special Funding Program for Graduate Student Innovation in Jiangxi Province, grant number YC2023-B140 (H. L.). The above funders all have contirbuted to this study, the roles they took in the study are as followings: H. L. (First Author): Conceptualization, Methodology, Software, Formal Analysis, Writing—Original Draft; Q. L.: Formal Analysis,Resources,Funding acquisition B. L.: Validation, Supervision, Writing—Review & editing, Funding acquisition Z.X. C.: Validation, Supervision, Writing—Review & editing, Funding acquisition Y.M. Y.: Software, Resources In view of the fact that B. L. and Z.X. C. shared the supervision of this article, the two are designated as co-corresponding authors.

**Competing interests:** The authors have declared that no competing interests exist.

shortages and escalating factor costs have led to a "crowding effect" in the coastal areas of China, causing the manufacturing industry to gradually relocate toward the central and western regions [2].

Agro-processing is an important industry in the national economy of China. It not only serves as a vital link connecting the primary, secondary, and tertiary industries, but also plays a key role in increasing the added value of agricultural products and constructing the agricultural industry chain. The Opinions of the CPC Central Committee and the State Council on Key Tasks for Promoting Rural Vitalization in 2023 advocates upgrading the agro-processing industry; supporting the development of initial processing of agricultural products at family farms, farmers' cooperatives, and small and medium-sized agricultural production enterprises; and guiding large agricultural enterprises to develop deep processing of their products. With the Chinese government at all levels paying increasing attention to agriculture, the agro-processing industry has begun vigorous development, and many agglomeration areas for agro-processing have emerged, such as the Guangdong food processing and manufacturing industry, the Yunnan tobacco products industry, the Jiangsu textile industry, and the Zhejiang clothing and footwear industry. In 2022, it was estimated that there will be 90,000 above-scale agricultural product processing enterprises in China, with such enterprises achieving a total operating income of more than RMB 18.5 trillion. The conversion rate of agricultural product processing will reach 72%, and the ratio of the output value of agricultural product processing to that of agriculture will reach 2.52:1 [3].

From the academic point of view, studies on agro-processing industry agglomeration mainly focus on the following aspects. First, some studies adopted different indicators to measure the degree of industrial agglomeration including H-index, spatial Gini coefficient, average concentration ratio, industrial center value, K function, Moran coefficient [4–8]. Qin et al. [9], Su et al. [10], Deng et al. [11], Feng et al. [12], He et al. [13] analyzed industrial agglomeration from different spatial scales. In addition, Scott [14], Lambert et al. [15]), Wu et al. [16]), Zhou et al. [17], Xia et al. [18]) and Zhao et al. [19] explored the degree of industrial agglomeration from different industries, such as corn processing industry, feed industry, wood processing industry and garment manufacturing industry. Second, some researchers, such as Huang et al. [20]) and Jia et al. [21] concluded that agro-processing industry agglomeration has a strong positive correlation with agricultural economic growth. Ping et al. [22] found that the driving effect of industrial agglomeration on agricultural economy show an "inverted u-shaped" relationship in the Yangtze River Delta region. While Xu et al. [23]) and Zeng et al. [24] found that the industrial agglomeration may have a negative effect on production efficiency. Moreover, Cheng [25] and Wang et al. [26] studied the relationship between manufacturing agglomeration and environmental pollution. Third, the literatures addressing the change mechanism of agro-processing industry agglomeration revealed that the natural endowments, geographic location, human capital endowments, transaction costs, market demand, infrastructure, openness policies, financial development, policy intervention are the main motivations for promoting change in the agro-processing layout [27, 28].

In summary, first, although some researchers have used different research methods to analyze the changes of agro-processing industry agglomeration from different spatial scales, the conclusions of these studies are quite different due to the differences in research intervals, methods, and measurement indexes. And it is difficult to reflect the evolution of the pattern of China's agro-processing industry and the characteristics of the agglomeration in the last 20 years. Second, the criteria for dividing the agro-processing industry have yet to be explored. For example, most researchers have included the "printing and recording media reproduction industry" in the scope of agro-processing industry, but this industry is not very relevant to agriculture and cannot reflect the characteristics of agro-processing. Third, the impact of

production factors input on industrial agglomeration has been addressed in some studies, but the descriptions are relatively general and broad, and there is a lack of quantitative research on production factors input and the degree of industrial agglomeration.

How did the agro-processing industry agglomeration change in the last two decades? Were the production factors inputs relevant to the industrial agglomeration? Did they play a crowding effect or a inducing effect? Therefore, this study observed dynamic changes of agro-processing industry agglomeration based on provincial panel data of China from 2001 to 2020. Then a theoretical analysis framework was established and a dynamic spatial Durbin model was used to quantitatively explore the spatial effects of production factors on the industrial agglomeration. Based on foregoing analysis, it provides a reference basis for the government to optimize the agro-processing industry spatial layout by adjusting production factor inputs as well as promote high-quality development of the agro-processing industry.

The content of the paper is as follows: The first section summarizes the research progress and puts forward the research hypotheses. The second section discusses the model construction and data indicators. The spatial regression results and discussions are presented in the third section, while the main conclusions and policy relevance are presented in the fourth section.

## Theoretical analysis

Agglomeration is an important phenomenon that accompanies economic development. Thunen [29], Launhardt [30], Weber [31], and others studied the location selection of agriculture and industry from the perspective of transportation costs. Christaller [32] believed that in addition to considering production costs, demand boundaries and market scope need to be taken into account when locating a business. Marshall [33] and Krugman [34], on the other hand, attempted to find explanations for industrial agglomeration based on endogenous economic factors. Porter [35] supposed that it is the intense competition brought about by the concentration of numerous small and medium-sized enterprises in an area that generates innovation and reformation, and this is one of the crucial ways to fuel the competitiveness of a country or region. Many scholars have considered the formation mechanisms and driving factors of industrial agglomeration, while others have observed the dynamic changes of agglomeration. Hoover [36] proposed the theory of optimal size for industrial agglomeration, suggesting that the number of firms concentrated in a certain location cannot be too small, otherwise the best effects of agglomeration cannot be achieved. Conversely, having too many firms in the same location will lower the overall positive effects of agglomeration. Brakman and others [37] were the first to propose the concept of an agglomeration "crowding effect", suggesting that as urban areas expand in size, the negative effects of crowding will outweigh the positive effects of agglomeration. Tichy [38] and Maggioni [39] analyzed the birth, growth, maturity, and decline of industrial agglomeration from the perspective of life cycle theory, and proposed that agglomeration in the decline stage would lead to diseconomies of scale.

In general, traditional production factors include land (natural resources), capital, and labor. With the popularization of informatization and intelligentization in modern society, knowledge and experience accumulated in production and life have become a new type of production factor widely used by enterprises. Therefore, this article focused on capital, labor and technology production factors (As land is an exogenous given and the stock of natural resources is fixed, especially since the quantity of manufacturing industries does not change much, in this article we did not include land in the framework of factor endowment research.).

According to the theory of new structural economics, the factor endowments of an economy are given at each time point, and the economic structure is endogenous to the factor

endowment structure [40, 41]. When labor resources are abundant and capital resources are scarce, the focus should be on developing labor-intensive industries. With an upgraded factor endowment structure, when capital becomes sufficient, the industrial structure should be gradually upgraded to capital-intensive industries. When technology and other factors become comparative advantages, the dominant industries should shift toward technology-intensive ones. Hence, the structure of factor endowments and the efficiency of their allocation in a region are closely related to the spatial distribution of industries [42], which directly leads to the concentration or dispersion of industrial activities. In particular, the combination of factors such as capital, labor, and knowledge are key in determining the industry location [43, 44]. Based on the above analysis, the following hypotheses was made:

**H1:** The input of capital, labor, and technology production factors will affect the choice of location for agro-processing industry.

While agglomeration generates economies of scale, the increasing return will only enhance the existing locational advantage to a certain extent. With the constant expansion of industry scale and fierce competition among enterprises, the proportion and relative prices of factors will change, which will produce a negative effect of industrial agglomeration as "crowding effect" [45, 46]. This refers to the non-economic consequences resulting from the imbalanced allocation of production factors caused by economic over-agglomeration within a certain spatial range [47]. Tang et al. [48], Wang and Tang [49], Shen et al. [50], and Zeng et al. [51] used data envelopment analysis to verify the whole effect of "factor congestion" in the manufacturing industry in some regions of China. However, they did not specifically elaborate on the individual effects of each production factor.

Marx pointed out that "Capital itself always appears as a value that will directly increase on its own." The appreciation characteristic of capital drives a constant pursuit for maximization of profits, with the aim of achieving maximum output with minimal input and continuously shifting from low-productivity to high-productivity industries. There is a strong profit-seeking tendency [52]. Therefore, this study proposed hypothesis:

**H2:** The input of capital factors has an inducing effect on agglomeration in the agro-processing industry.

Generally, the labor factor flows to regions with higher actual rewards, which depends not only on current income differences between two regions, but also on future factor rewards in the regions. Especially, people's "voting with their feet" is determined by their expectation of future factor rewards, which means they will move to a region they think is promising until it becomes the agglomeration center [53]. However, some scholars have found in their research that the inducing effect of the labor factor gradually weakens after the development of industrial agglomeration reaches a certain level. For example, Galarraga et al. [54] analyzed the relationship between the spatial density of economic activity and industrial labor productivity in Spain from 1860 to 1999, and found that a crowding effect occurred in the last period of the 20th century, which caused the dispersion of local industrial activities. Ke and Yao [55], using Chinese cities at the prefecture level and above as the research object, determined that the spatial density of employment in Chinese cities was excessively high and the crowding effect of industrial agglomeration would lead to a decrease in labor productivity. Yan and Qiao [56], in their study on the agglomeration of Chinese manufacturing, discovered that the agglomeration of low-tech and labor-intensive industries had already produced a labor crowding effect. Zhou and Zhu [47] also claimed that since 2003, the constraint effect on the industry of labor crowding caused by the labor factor has become dramatically prominent in China. Chen and Miao [57] measured the degree of congestion of manufacturing production factors in the Yangtze River Delta region and found that there is a redundancy of labor factors in the textile industry. Based on the conclusions of the studies mentioned above, we propose hypothesis:

**H3:** The input of labor factors has a crowding effect on the agglomeration of the agro-processing industry.

It is generally believed that the rapid development of scientific and technological information has led to the implementation of remote administration and supply chain management, which has reduced the cost of finding trading partners and the time required for long-distance contacts. This "disappearance of the distance effect" weakens the justification for the existence of industrial agglomeration and promotes the dispersion of industries [53]. Additionally, some scholars have put forward a different view, that the spike in information volume and the deepening of information complexity have actually facilitated more close-range communication, because only enterprises that are spatially agglomerated can effectively obtain the tacit knowledge brought about by such close-range communication [58]. When studying the agglomeration of high-tech manufacturing industries, Sedgley and Elmslie [59] found that for every standard deviation of 1 added to the location quotient compared to the mean value, the average patent ownership per 10,000 workers would increase by 1.7 times. Chen and Miao [57], in their research, found that the knowledge spillover level has a significant promoting effect on spatial agglomeration in China. Therefore, we propose hypothesis:

**H4:** There is uncertainty in the effects of technological factor input.

## Model construction and data sources

First, the dynamic changes of agro-processing industry of 29 provinces in China was described by calculating the spatial Gini coefficient as well as the average concentration ratio. Second, this section verified the spatial effects of production factors on China's agro-processing industry including capital, labor and technology factors, by constructing a dynamic spatial Durbin model.

### Empirical model construction

**Industry spatial Gini coefficient.**   Krugman [34] introduced the spatial Gini coefficient for measuring the degree of distributional equity among industry regions using the Lorenz curve. The frequently used spatial Gini coefficient nowadays is the relative Gini coefficient, which is generally calculated using Formula (1) [60]:

$$G = \left( 2 \times \sum_{i=1}^{n} \sum_{j=1}^{m} \left| s_{ki} - s_{kj} \right| \right) / 2n^2 s_k \tag{1}$$

where $n$ represents the number of regions, $s_{ki}$ and $s_{kj}$ represent the share of industry $k$ in regions $i$ and $j$, and $s_k$ represents the average proportion of industry $k$ in each region. When an industry is evenly distributed across all regions, the $G$ value is 0, and when an industry is completely concentrated in one region, the $G$ value is 1. An increase in the Gini coefficient implies an increase in the concentration of industries.

**Regional average concentration ratio.**   The regional average concentration ratio index in Formulas (2) and (3) can measure the average market share of various industries in a certain region compared to the national market [61]. The range of the index is from 0 to 1, and values closer to 1 indicate a higher average market share of the industry in the region, indicating a more developed industry. In the equations, $i$ and $k$ represent region $i$ and industry $k$, and $E_i^k$

indicates the total output value of industry $k$ in region $i$.

$$v_i = \sum_k (v_i^k)/k \qquad (2)$$

$$v_i^k = E_i^k / \sum_i E_i^k \qquad (3)$$

**Dynamic spatial Durbin model.** Owing to the complex spatial relationships in industrial agglomeration within a region, as well as the mobility of factors such as capital, labor, and technology, a factor in one region will affect the industrial agglomeration not only in that region but also in neighboring regions. According to the first law of geography, everything is related to everything else, and things that closer together are more related than things that are farther apart. This study adopts spatial econometric models as the main empirical method and constructs a spatial weight matrix based on geographical distance to analyze the influencing factors and spillover effects of agro-processing agglomeration. This article uses the direct distance ($d_{ij}$) between the two provincial capital cities as the standard, and the smaller the distance between two cities, the larger the spatial weight of the geographical distance, as shown in Formula (4):

$$W_{ij} = \begin{cases} 1/d_{ij}, & (i \neq j) \\ 0, & (i = j) \end{cases} \qquad (4)$$

where $W_{ij}$ represents the spatial weight matrix between province $i$ and province $j$, and $d_{ij}$ represents the straight-line distance between the provincial capitals of province $i$ and province $j$. When $i = j$, $W_{ij}$ takes a value of 0.

**Model construction.** Traditional spatial econometric models mainly include spatial lag models (SLMs) and spatial error models (SEMs). The spatial Durbin model (SDM) combines these two types of models and effectively reduces the bias in spatial autoregressive coefficients, thereby highly improving the explanatory power of the model. At the same time, taking endogeneity issues into account, in this paper we use a dynamic spatial Durbin model as an empirical method to study the agglomeration factors of the agro-processing industry:

$$INA_{i,t} = \lambda_1 INA_{i,t-1} + \lambda_2 WINA_{i,t-1} + \lambda_3 WINA_{i,t} + \lambda_4 X_{i,t} + \lambda_5 WX_{i,t} + \mu_i + v_i + \epsilon_{i,t} \qquad (5)$$

In Eq (5), $i$ represents provinces, $t$ represents years, and $W$ is the spatial geographic distance weight matrix; $INA_{i,t}$ and $INA_{i,t-1}$ represent the agglomeration degree of agro-processing in province $i$ in years $t$ and $t-1$, respectively; $WINA_{i,t}$ and $WINA_{i,t-1}$ represent the agglomeration degree of agro-processing in provinces adjacent to province $i$ in years $t$ and $t-1$, respectively; $X_{it}$ is the matrix of explanatory and control variables ($X = CAP, LAB, TEC, GOV, FIN, FDI, TRA, RES, INF, OPE, URB$); $\lambda 1–\lambda 5$ are the structural parameters of the econometric model, which describe the determinants of industrial agglomeration in terms of the orientation and strength of their relationship with the degree of agglomeration; $\mu_i$ represents individual fixed effects; $v_i$ represents fixed time effects; and $\varepsilon_{i,t}$ represents the random disturbance term.

## Variable selection

**Dependent variable.** The dependent variable is the industrial agglomeration index *(INA)*. When using the location quotient to examine the status of regional industrial agglomeration, the higher the location quotient, the greater the degree of agglomeration of a certain industry in one region, and the natural logarithm is inserted into the model. The formula for using the

location quotient to calculate industrial agglomeration index is as follows:

$$INA_{tij} = \left( e_{tij}/E_{ti} \right) / \left( e_{tj}/E_t \right) \tag{6}$$

In Eq (6), $e_{tij}$ represents the gross product value of industry $j$ in province $i$ in year $t$, $E_{ti}$ represents the gross product value of province $i$ in year $t$, $e_{tj}$ represents the gross product value of industry $j$ in year $t$ in the country, and $E_t$ represents the gross product value of the country in year $t$.

**Independent variables.**   Capital input ($CAP$) is measured by the proportion of the total amount of fixed capital formation to gross domestic product [62], and the natural logarithm is taken into the model. Labor input ($LAB$) is expressed as the ratio of the number of employees in the secondary industry to the total number of employed persons, and the natural logarithm is taken into the model. Technological input ($TEC$) is expressed as the ratio of total R&D spending to GDP [63], taking the natural logarithm into the model.

**Control variables.**   Although in this paper we use a two-way fixed effects model and lag both the explanatory and core explanatory variables one period to avoid endogeneity issues in the model, it is still possible to omit the estimation bias caused by variables. Therefore, drawing on existing literature [53, 63–67], we control for the following variables.

Government intervention level ($GOV$): Industry agglomeration and diffusion are inevitably affected by policies, and the degree of government intervention is measured by the ratio of general budget expenditure to GDP.

Financial development level ($FIN$): The development of modern industries cannot lack financial support. The impact of the financial sector is represented by the proportion of value added of the financial sector to GDP, and the natural logarithm is taken into the model.

Foreign direct investment ($FDI$): Foreign direct investment is a salient spatial agglomeration phenomenon, and therefore directly affects the concentration and dispersion of the industry. In the model, foreign direct investment is taken as a natural logarithm.

Transportation infrastructure ($TRA$): Transportation costs affect agglomeration through industrial location, and transportation infrastructure is a vital factor affecting transportation costs. As highways account for more than 85% of total societal freight volume, in this paper we use highway mileage to represent transportation infrastructure and incorporate the natural logarithm into the model.

Natural resource endowment ($RES$): The location of an industry depends on the interaction between the proximity of consumer, supply, and factor markets. Therefore, the agricultural resource endowment directly affects the agglomeration degree of industries. In this paper we use the proportion of the total output value of agriculture, forestry, animal husbandry, and fishery to GDP to represent this variable.

Information infrastructure ($INF$): With the development and rise of information technology, information infrastructure promotes the circulation of production factors such as capital and human resources, and balances economic agglomeration development. The ratio of the length of long-distance fiber optic cable to the land area is used to measure information infrastructure, and the natural logarithm is used in the model.

Degree of openness to the outside world ($OPE$): Regions with more foreign investment also have more exports. The indicator of degree of openness to the outside world affects industrial agglomeration as well. It is represented by the proportion of total export–import volume to GDP.

Level of urbanization development ($URB$): Generally speaking, industry agglomeration will lead to population agglomeration at the same time. The urbanization rate is adopted to represent the level of urbanization development.

### Data sources and processing

According to the Chinese National Economic Industry Classification (*GB/T4754-2017*), the judging criterion is whether agricultural resources are directly used as production inputs. The scope of agro-processing in this article is defined as follows: farm and sideline food processing industry (*C13*); food manufacturing industry (*C14*); wine, beverage, and refined tea manufacturing industry (*C15*); tobacco manufacturing industry (*C16*); textile industry (*C17*); textile, garment, and accessory manufacturing industry (*C18*); leather, fur, feathers, and their products and footwear manufacturing industry (*C19*); wood processing and wood, bamboo, rattan, palm, and straw products industry (*C20*); furniture manufacturing industry (*C21*); and papermaking and paper products industry (*C22*). Additionally, the study includes the processing of traditional Chinese medicine decoctions and the manufacture of Chinese patent medicines in the pharmaceutical manufacturing industry, which are combined into the Chinese medicine manufacturing industry (*C27*), making a total of 11 sub-industries.

The gross industrial production data of agro-processing in this paper mainly comes from the China Industrial Economic Statistics Yearbook (2001–2020) (Due an adjustment in the threshold for the surveyed enterprises in 2001 from an annual main business income of RMB 5 million to RMB 20 million, in order to ensure consistency in the "above-scale" standard of the selected samples in each year to the greatest extent possible, we set the research period to 2001–2020.), China Industrial Enterprise Database (2001–2015), and China's four economic censuses (2004–2018). However, data on various indicators for Tibet Autonomous Region and Qinghai Province are relatively sparse, so we did not include them in the research scope of this study. Other data are from the China Statistical Yearbook, China Science and Technology Statistics Yearbook, China Labor Statistical Yearbook, China Population and Employment Statistics Yearbook, and various provincial statistical yearbooks from relevant years. Some missing values were estimated by linear interpolation.

## Empirical results and analysis

This section described the dynamic changes of China's agro-processing industry agglomeration from 2001 to 2020 from different industries and regions. In addition, the spatial spillover effects of production factors such as capital, labor, and technology on industrial agglomeration were specifically examined from short-term and long-term perspectives.

### Current status of Industrial agglomeration

**Agglomeration status of different industries.**   Using Formula (1), we calculated the agglomeration level of the agro-processing industry. As shown in Fig 1, from 2001 to 2020, the agglomeration level of China's agro-processing, primary processing, and deep processing industries all showed an overall downward trend, with respective decreases of 7.02%, 14.43%, and 1.86%. At the same time, it exhibited a fluctuating pattern of "rise–fall–rise–fall" and reached peaks in 2006 and 2016 respectively. This may have been influenced by industrial policies in those years. For example, in 2016, the Chinese Ministry of Agriculture and Rural Affairs issued the National Plan for the Integrated Development of Agro-Processing Industry and the First, Second, and Third Industries in Rural Areas (2016–2020), which may have had an impact on the agglomeration level of agro-processing. In terms of processing classification, the agglomeration level of deep processing industry has always been higher than the average level of primary processing industry, and this discrepancy has been constantly changing over time. Before 2015, the gap gradually narrowed, but after 2015, it began to widen again.

The above conclusion regarding the changes in agglomeration of agro-processing is exactly opposite to the trend of industrial agglomeration from the early days of reform and opening

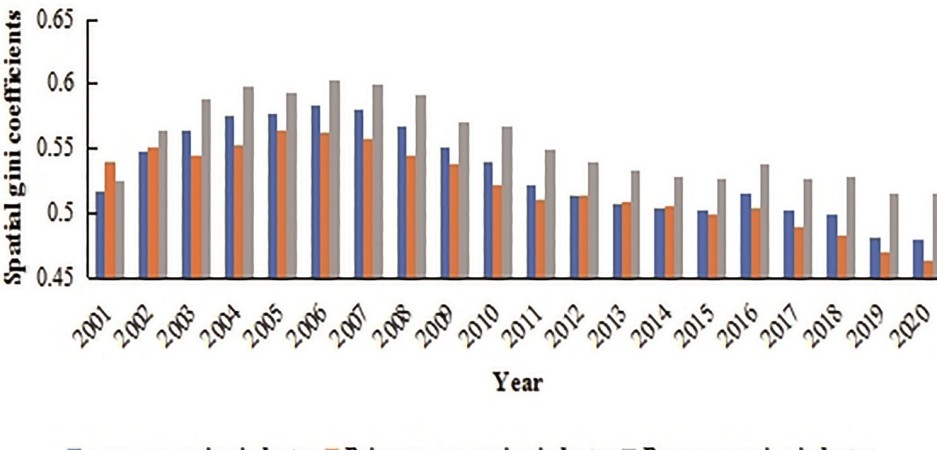

**Fig 1. Spatial Gini coefficient of China's agro-processing industry and sub-industries (2001–2020).** Source: China Industrial Economic Statistics Yearbook, China Industrial Enterprise Database and China's four economic censuses.

up to 2000 [61, 68]. Overall, the changes in agglomeration of China's ago-processing industry have been closely related to the development strategies of each stage since the 1980s. The ascending stage of industrial agglomeration coincided with the rapid development period of the eastern coastal regions, aided by the "spring breeze" of reform and opening up. The descending stage was the period when China's manufacturing industry gradually shifted from the eastern coastal regions to the central and western regions. This can be further verified and elaborated by analyzing the agglomeration situation in different regions.

**Agglomeration status in different regions.** China can be divided into six regions based on the geographic environment and economic development: municipalities directly under the central government, eastern coastal areas, northeast China, central areas, northwestern areas,

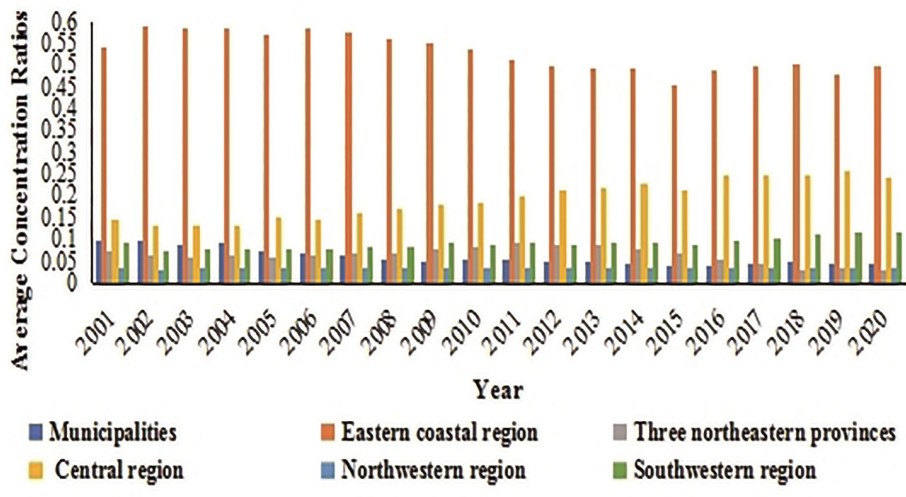

**Fig 2. Comparison of average concentration ratio of agro-processing industry in different regions of China (2001 to 2020).** Source: China Industrial Economic Statistics Yearbook, China Industrial Enterprise Database and China's four economic censuses.

and southwestern areas (Municipalities include Beijing, Shanghai, and Tianjin. Eastern coastal region include Hebei, Shandong, Jiangsu, Zhejiang, Fujian, Guangdong, Guangxi, and Hainan. Northeastern provinces include Heilongjiang, Jilin, and Liaoning. Central region include Shanxi, Henan, Anhui, Hubei, Hunan, and Jiangxi. Northwestern region include Inner Mongolia, Shaanxi, Ningxia, Gansu, and Xinjiang. Southwestern region include Sichuan, Chongqing, Yunnan, and Guizhou.). We can conduct a comparative analysis on the degree of agglomeration and its changes in each region. By using Eqs (2) and (3), the average concentration ratio of agro-processing can be obtained for each region (Fig 2).

After more than 20 years of development, the distribution of China's agro-processing has undergone shifts and adjustments. The average concentration ratio of the eastern coastal areas, which used to have a geographical advantage, decreased from 0.5432 in 2001 to 0.4990 in 2020, a drop of 8.14%. The average concentration ratio of municipalities directly under the central government and in the northeastern region also declined, with drops of 52.56% and 55.48%, respectively. The northwestern and southwestern regions had relatively low average concentration ratios in 2001; however, over time, there has been an obvious upward trend, especially in the southwestern region, with an increase of up to 30%. Meanwhile, the attractiveness of the central region for the agro-processing industry gradually increased, with the average concentration ratio rising from 0.1489 in 2001 to 0.2424 in 2020, an increase of 62.87%.

## Empirical study

### Spatial econometric model testing

First, the global Moran's I index was used to test the existence of spatial autocorrelation in the model. The results indicate that the Moran's I of the agglomeration degree of agro-processing is significant at the 1% level, which means that the data are spatially autocorrelated (Table 1). Additionally, the results of calculating the Lagrange multiplier (LM) and robust statistics evidently reject the original hypothesis of no spatial correlation, indicating that using spatial econometric methods is reasonable (Table 1). Subsequently, the LR test was conducted on the dynamic spatial Durbin model of industrial agglomeration, and the results show that this model could be used for empirical analysis (Table 2). The Hausman test was conducted on the dynamic spatial Durbin model, and the findings show that the fixed effects model should be used to analyze the factors influencing the agglomeration of the agro-processing industry (Table 2). After controlling time fixed effects, individual fixed effects, and bidirectional fixed effects, it is recommended to use bidirectional fixed effects for examination (Table 3).

**Discussion of results.** Using Stata 16 software, we conducted an empirical analysis of the elasticity coefficients and spatial spillover effects of production factor on industrial agglomeration, as well as other influencing factors, based on the dynamic spatial Durbin model with bidirectional fixed effects.

*Production factor inputs and industrial agglomeration.* Based on the results in Table 4, both the time lag effect and the dual spatiotemporal effects of agglomeration in agro-processing are noteworthy at the 1% level, indicating that agglomeration has an inertial effect. Once a region becomes an agglomeration area for agro-processing, there will continue to be a trend of agglomeration development, and regions with close geographic ties can also develop into agglomeration areas.

Furthermore, capital, labor, and technology inputs have an overt promoting effect on agglomeration in agro-processing at the 1% level, showing an inducing effect, but with different degrees of impact. Regions with higher inputs of labor, capital, and technology are more likely to attract and develop agglomerations in agro-processing. Among them, the positive promotion effect of labor input is the strongest, with a 1% investment boost promoting a

**Table 1. Results of Moran's index, LM test, and robustness statistics for agglomeration degree of China's agro-processing industry.**

| Industry | Tests | | Statistics | Freedom | p-value |
|---|---|---|---|---|---|
| Agro-processing | Spatial error | Moran's I | 30.523 | 1 | 0.0000 |
| | | LM test | 840.358 | 1 | 0.0000 |
| | | Robust LM test | 351.232 | 1 | 0.0000 |
| | Spatial lag | LM test | 667.002 | 1 | 0.0000 |
| | | Robust LM test | 177.877 | 1 | 0.0000 |

2.212% increase in agglomeration. The second is capital input, with a regression coefficient of 0.614. The impact of technology factor input on industrial agglomeration is the weakest, with a regression coefficient of 0.028, which indirectly demonstrates that there is great potential for technology factors to play a role in the inducing effect. The effect of labor factor input is contradictory to the previous hypothesis, mainly because this hypothesis was based on some conclusions in the literature that were based on research of industrial manufacturing as a whole. This also indicates that there are apparent divergences in the distribution and flow of labor across assorted manufacturing sub-industries. For example, some high-tech and high-end manufacturing industries have already experienced a crowding effect in terms of labor supply, but the demand for labor in the agro-processing industry remains high. Therefore, the development of the agricultural industry still requires continuous investment in human capital.

The endowment of agricultural resources, the level of urban development, the degree of government intervention, foreign direct investment, and information infrastructure have a significant promotion effect on the agro-processing industry. In other words, regions that are closer to the source of raw materials and have higher levels of urbanization, greater government fiscal investment, more foreign direct investment, and more advanced information infrastructure are more likely to form clusters of agro-processing industries. The level of financial development has a certain negative impact on agro-processing industry agglomeration, which reflects the low availability of financial loans for agricultural enterprises. The degree of trade openness has a strong negative effect on industrial agglomeration at the 1% level, indicating that China's processing products of agriculture is still mainly for domestic consumption and does not have a remarkable advantage in export trade.

*Spatial spillover effect.* In order to further illustrate the spatial effects among variables, we also calculated the direct and indirect effects of the dynamic spatial Durbin model in the short and long term.

**1. Short-term spatial effects (Table 5).** On the one hand, the direct effects in the short term show that the input of capital, labor, and technology still has an inducing effect on agglomeration of local agro-processing. Additionally, it is positively influenced by government intervention, foreign investment, natural endowments of agriculture, the level of information infrastructure, and urban development. On the other hand, the indirect effects in the short term indicate that increasing the input of production factors in neighboring areas will also promote agglomeration of local agro-processing. Furthermore, increasing financial input in neighboring regions, expanding foreign investment, developing agricultural production, and

**Table 2. LR, Hausman, and significance tests for agglomeration degree of China's agro-processing industry.**

| Industry | Tests | | LR chi$^2$(10) | p-value |
|---|---|---|---|---|
| Agro-processing | LR test | SDM and SEM tests | 35.38 | 0.0001 |
| | | SDM and SAR tests | 34.78 | 0.0001 |
| | | Hausman test | 307.11 | 0.0000 |

**Table 3. Test results of bidirectional fixed effects for agglomeration degree of China's agro-processing industry.**

| Industry | Tests | LR chi$^2$(10) | p-value |
|---|---|---|---|
| Agro-processing | Ind and both tests | 99.71 | 0.0001 |
| | Time and both tests | 599.97 | 0.0001 |

increasing the level of urbanization have spillover effects, which will have a positive impact on the growth of the local agro-processing industry.

**2. Long-term spatial effects (Table 6).** Although the production factor inputs in the local area still plays a promoting role in industrial agglomeration, the siphon effect produced by the production factors in neighboring areas will offset the positive effects, resulting in a negative total spatial effect. Therefore, the inputs of labor, capital, and technology factors will exhibit a crowding effect in the long run, with the degree of crowding decreasing in sequence. Additionally, government intervention has a promoting effect in the initial stages of industrial development, but for enterprises that do not have endogenous capability in the later stages, it may have an inhibitory effect. Massive foreign investment, agricultural resources, and urban population can also lead to an acute crowding phenomenon. On the contrary, the total long-term effects of the level of financial industry development, transportation infrastructure construction, and foreign trade are all significant at the 1% level. This shows that these influencing factors will further promote the sustaining growth of agro-processing in the long term.

## Heteroscedasticity test

Because of the differences in geographic location, agglomeration characteristics, and agglomeration forms among agro-processing sub-industries, the promoting effects of production factor inputs on agglomeration can also exhibit spatial heterogeneity among sub-industries. Therefore, a spatial econometric test based on sub-industries is necessary.

We verified the industry heterogeneity of agro-processing by dividing it into two types, primary and deep processing, based on the degree of change that occurs in processed agricultural products. Primary processing generally refers to changes in the quantity of processed agricultural products and does not involve changes in the internal structure or physical and chemical properties, which mainly refers to farm and sideline food processing (C13) in this article. Deep processing generally involves qualitative changes, such as extracting and transforming nutrients, including proteins, fats, fibers, etc., and active ingredients in agricultural products. This includes all other sub-industries in agro-processing except for farm and sideline food processing (C13). To verify the heterogeneity of the samples in the study, a dynamic SDM model of agglomeration under a spatial geographic matrix was calculated for both the primary and deep agro-processing industries.

It can be seen in S1 Table that capital factor input has a positive effect on the agglomeration of both the primary and deep processing industries, with a more conspicuous effect on the former. An increase of 1% in capital input will lead to a growth in agglomeration of 1.29% and 0.66%, respectively. The direction of the effect of labor factor input on the two industries is negative. There is a crowding effect in the primary processing industry, and an inducing effect in the deep processing industry. A tremendous influx of low-skilled labor into the primary processing industry could lead to excessive labor factor input, while the high-skilled labor resources required to match the posts in the deep processing industry are still lacking. The effect of technological factor input on agglomeration in the primary processing industry is not apparent, but it plays a significant role in promoting agglomeration in the deep processing industry, indicating that the overall level of technological innovation in the primary processing

**Table 4. Influence of production factor input on agglomeration of agro-processing industry in China (2001–2020).**

| Variables | Main effects | | Spatial spillover effects |
|---|---|---|---|
| Time-lag effect | 0.895*** | | |
| | (18.792) | | |
| Dual spatiotemporal lag effect | 1.015*** | | |
| | (3.347) | | |
| CAP | 0.614*** | WxCAP | 0.862* |
| | (6.364) | | (1.817) |
| LAB | 2.212*** | WxLAB | 57.059*** |
| | (7.619) | | (42.254) |
| TEC | 0.028*** | WxTEC | 0.113*** |
| | (10.398) | | (6.545) |
| GOV | 3.038*** | WxGOV | 41.953*** |
| | (9.277) | | (20.123) |
| FIN | -0.113*** | WxFIN | -4.234*** |
| | (-2.729) | | (-14.636) |
| FDI | 0.043*** | WxFDI | 0.815*** |
| | (3.004) | | (8.843) |
| TRA | -0.029 | Wxtra | -4.336*** |
| | (-0.591) | | (-10.624) |
| RES | 9.417*** | WxRES | 94.343*** |
| | (26.245) | | (47.790) |
| INF | 0.030*** | WxINF | -0.050 |
| | (4.757) | | (-0.973) |
| OPE | -0.852*** | WxOPE | -3.151*** |
| | (-16.135) | | (-13.512) |
| URB | 4.147*** | WxURB | 28.164*** |
| | | | (16.413) |
| Observations | 551 | | 551 |
| R2 | 0.421 | | 0.421 |

Note: *, **, and ***indicate significance at 10%, 5% and 1% levels, respectively. Values in parentheses are z-statistics.

industry is not high, and the role of knowledge and technology in promoting its revitalization and development needs to be functionalized. Moreover, whether short term or long term (S2 Table), except for the alternation in the direction of the labor and technology effect of the primary processing industry, the signs of the other variables are basically consistent with the previous results of agro-processing (Table 4), which partly solves the heterogeneity problem in this article.

## Robustness test

**Spatial econometric test with shorter time windows.** In 2016, the Chinese Ministry of Agriculture and Rural Affairs issued the National Plan for the Integrated Development of Agricultural Processing Industry and the First, Second, and Third Industries in Rural Areas (2016–2020), which may have had an impact on the pattern and agglomeration level of agro-processing (Figs 1 and 2). To further exclude the influence of external policies and avoid confounding of the empirical results after the complementary treatment of the 2017 data, a smaller sample (from 2001 to 2015) was used for estimation, in order to verify the robustness of the model. As shown in S3 Table, the coefficient signs and the significance of labor and technology factors

Table 5. Short-term spatial effects of production factor input on industrial agglomeration of China's agro-processing industry.

| Variables | Direct short-term effects | Indirect short-term effects | Total short-term effects |
|---|---|---|---|
| CAP | 0.622*** | 0.852* | 1.473*** |
| | (6.764) | (1.706) | (2.892) |
| LAB | 2.147*** | 56.458*** | 58.605*** |
| | (3.541) | (5.745) | (5.651) |
| TEC | 0.028*** | 0.112*** | 0.141*** |
| | (10.377) | (3.657) | (4.443) |
| GOV | 3.012*** | 41.703*** | 44.715*** |
| | (5.715) | (4.874) | (4.992) |
| FIN | -0.110* | -4.208*** | -4.318*** |
| | (-1.885) | (-5.109) | (-5.004) |
| FDI | 0.042*** | 0.803*** | 0.845*** |
| | (3.351) | (4.969) | (5.156) |
| TRA | -0.022 | -4.279*** | -4.302*** |
| | (-0.313) | (-5.644) | (-5.358) |
| RES | 9.341*** | 93.325*** | 102.666*** |
| | (10.287) | (5.559) | (5.816) |
| INF | 0.031*** | -0.046 | -0.016 |
| | (4.836) | (-0.930) | (-0.296) |
| OPE | -0.849*** | -3.119*** | -3.968*** |
| | (-12.901) | (-4.350) | (-5.298) |
| URB | 4.110*** | 27.947*** | 32.058*** |
| | (10.503) | (5.358) | (5.832) |
| Observations | 551 | 551 | 551 |
| R2 | 0.421 | 0.421 | 0.421 |

Note: *, **, and *** indicate significance at 10%, 5%, and 1% levels, respectively. Values in parentheses are z-statistics.

are approximately consistent with the data in Tables 4–6, except for the insignificant effect of the capital factor.

**Spatial econometric test based on dependent variables.** In this paper we also used the method of replacing the dependent variable to verify the robustness of the model. Industrial agglomeration is characterized by the average concentration ratio, and the natural logarithm is taken as the variable into the model, according to Eq (2). From the results in S4 Table, it can be found that the direction of the core explanatory variable is consistent with the results shown in Tables 4–6, which further verifies the robustness of the model in this paper.

## Conclusions and policy relevance

In this paper we used a dynamic spatial Durbin model of econometrics to conduct an empirical study of the relationship between production factor inputs and agglomeration of China's agro-processing, using panel data from 29 provinces from 2001 to 2020. The following conclusions are drawn: (1) the overall spatial agglomeration of China's agro-processing is constantly evolving as well as its sub-industries, showing a fluctuating of rise–fall–rise–fall. Looking at different industries, the deep processing industry has a distinctly higher agglomeration level than the primary processing industry, and this gap is constantly changing over time. Looking at different regions, the agglomeration level of industries in municipalities, the northeastern region, and the eastern coastal region all have declined, but the extent of the decline varies. Meanwhile,

**Table 6. Long-term spatial effects of production factor input on industrial agglomeration of China's agro-processing industry.**

| Variables | Direct long-term effect | Indirect long-term effect | Total long-term effect |
|---|---|---|---|
| CAP | 10.018 | -11.829 | -1.811** |
| | (0.058) | (-0.069) | (-2.418) |
| LAB | 67.776 | -138.536 | -70.760*** |
| | (0.033) | (-0.068) | (-4.638) |
| TEC | 0.492 | -0.662 | -0.169*** |
| | (0.052) | (-0.070) | (-4.045) |
| GOV | 67.149 | -120.763 | -53.614*** |
| | (0.039) | (-0.070) | (-5.031) |
| FIN | -3.383 | 8.583 | 5.199*** |
| | (-0.028) | (0.072) | (4.529) |
| FDI | 1.109 | -2.135 | -1.027*** |
| | (0.035) | (-0.068) | (-3.939) |
| TRA | -2.944 | 8.165 | 5.221*** |
| | (-0.024) | (0.067) | (4.034) |
| RES | 186.900 | -311.085 | -124.185*** |
| | (0.042) | (-0.070) | (-4.534) |
| INF | 0.364 | -0.336 | 0.028 |
| | (0.083) | (-0.077) | (0.419) |
| OPE | -13.378 | 18.155 | 4.777*** |
| | (-0.053) | (0.073) | (4.750) |
| URB | 74.096 | -112.981 | -38.885*** |
| | (0.046) | (-0.071) | (-4.277) |
| Observations | 551 | 551 | 551 |
| R2 | 0.421 | 0.421 | 0.421 |

Note: *, **, and ***indicate significance at 10%, 5%, and 1% levels, respectively. Values in parentheses are z-statistics.

the central and western regions have gradually become important areas for establishing and transferring agro-processing industries from the other regions, especially the central region, where their attractiveness is more prominent. (2) In the short term, the inputs of capital, labor, and technology factors have a significant promoting effect on the agglomeration of agro-processing industries in local and neighboring areas. Labor is the most influential factor, and technology is the least. In addition, the overall effect of labor and technological factors on the primary agricultural processing industry is negative, while capital factor has a positive effect. All three production factors have a positive total spatial effect on the deep processing industry. (3) In the long term, although the inputs of capital, labor, and technology in the local area still have a direct inducing effect on the agglomeration of the industry, the siphon effect generated by neighboring areas will counteract this effect, causing a crowding phenomenon in terms of total spatial effect. The total spatial effect of the three production factors on agglomeration of the primary processing industry is consistent with the short-term impact, but it has a negative total spatial effect on the deep processing industry.

Based on the above conclusions, we put forward corresponding policy recommendations in several areas including industrial planning, human capital investment, scientific and technological innovation, and new infrastructure construction.

1. The number and scale of agro-processing enterprises should be scientifically planned and controlled within a moderate range through rational intervention and active guidance. The

optimal proportion of production factors should be sought to fully leverage their comparative advantages, and excessive entry should be avoided. In the short term, preferential policies covering aspects such as land, funds, and talent can be focused on key cultivated enterprises in the early development stage of agro-processing. In the long term, the government should let the market play its role in efficiently allocating resources and cultivating the self-generating capability of enterprises, and gradually foster the competitive advantage of regional agro-processing through effective cooperative efforts between capable government and the private sector.

2. Investment in human capital should be expanded, and there should be a focus on popularizing basic education, improving vocational education, and utilizing multiple channels. The scientific literacy and skill level of the labor force should be reinforced, and high-quality labor should be guided to shift from the primary processing to the deep processing industry.

3. Research and development investment should be increased and the independent innovation capability of enterprises should be strengthened. Especially, efforts should be made to promote the upgrading of equipment and technological innovation in primary processing enterprises, accelerate production toward informatization and intelligentization, and further improve production efficiency.

4. There should be increased investment in new infrastructure, with the development of new infrastructure investment, taking information as the core, in parallel with investment in traditional infrastructure such as transportation. Adjacent areas should be encouraged to actively cooperate and reduce competition, facilitating positive spillover effects through complementary industrial division of labor.

## Supporting information

**S1 Table. Impact of production factor input on agglomeration of primary and deep agricultural processing industries in China.**
(DOCX)

**S2 Table. Short-term and long-term spatial effects of production factor input on agglomeration of primary and deep processing industries.**
(DOCX)

**S3 Table. Influence of production factor input on agglomeration of agro-processing industry in China (2001–2015).**
(DOCX)

**S4 Table. Impact of production factor input on average concentration ratio of China's agro-processing industry.**
(DOCX)

**S1 File.**
(DO)

**S2 File.**
(DTA)

## Acknowledgments

We extend our gratitude to Xinyi Tan and Ying Zhang, Liqing Zhu, Chao Feng and Yin Liu for their assistance in the field and for encouragement and support. We are also grateful to Zhen Wang and Xieqihua Liu. We particularly thank Xiaoyu Zhang, Dan Wu, Zhaoqing Cheng, who provided logistical support to facilitate our research. We would also like to thank Yongqi Yu for his support and help during the data analysis and all the participants of the writing workshop we attended for their constructive comments. Finally, we would like to thank the editors of PLOS ONE Journal and the two reviewers for their insightful comments.

## Author Contributions

**Conceptualization:** Hui Lu.

**Data curation:** Yongmei Ye.

**Formal analysis:** Hui Lu, Qing Li.

**Funding acquisition:** Hui Lu, Qing Li, Bin Liu, Zexin Chi.

**Methodology:** Hui Lu.

**Resources:** Qing Li.

**Software:** Hui Lu, Yongmei Ye.

**Supervision:** Bin Liu, Zexin Chi.

**Validation:** Bin Liu, Zexin Chi.

**Writing – original draft:** Hui Lu.

**Writing – review & editing:** Bin Liu, Zexin Chi.

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
