## [Decision Letter · Decision Letter 0]

31 Jul 2023

PONE-D-23-19856Research on the Dynamic Changes and Spatial Impact of Production Factors in China’s Agro-Processing Industry AgglomerationPLOS ONE

Dear Dr. Liu,

Thank you for submitting your manuscript to PLOS ONE. After careful consideration, we feel that it has merit but does not fully meet PLOS ONE’s publication criteria as it currently stands. Therefore, we invite you to submit a revised version of the manuscript that addresses the points raised during the review process.

Based on reviewers' comments and quality of the manuscript, you are required to make major modifications and improvements in the paper because it lacks clarity about research gap, novelty, a weak theoretical foundation, and the methodology employed does not sufficiently address the research questions, limiting its potential impact on the academic community. After incorporating all the changes and/or providing rebuttal statements where you have strong justification, kindly resubmit the manuscript  by responding to each comment raised for further evaluation.

We look forward to receiving your revised manuscript.

Kind regards,

Shujahat Haider Hashmi, PhD Regional Economics

Academic Editor

PLOS ONE

Journal Requirements:

"This research was funded by National Natural Science Foundation of China, grant number 17LZUJBWZX010; System of vegetable industrial technology of Jiangxi Province, grant number XARS-06; Project for key research items of humanities and social science for university in Jiangxi Province, grant number JD21080."

4. PLOS requires an ORCID iD for the corresponding author in Editorial Manager on papers submitted after December 6th, 2016. Please ensure that you have an ORCID iD and that it is validated in Editorial Manager. To do this, go to ‘Update my Information’ (in the upper center-hand corner of the main menu), and click on the Fetch/Validate link next to the ORCID field. This will take you to the ORCID site and allow you to create a new iD or authenticate a pre-existing iD in Editorial Manager. Please see the following video for instructions on linking an ORCID iD to your Editorial Manager account: https://www.youtube.com/watch?v=_xcclfuvtxQ

Additional Editor Comments:

Based on reviewers' comments and quality of the manuscript, you are required to make major modifications and improvements in the paper because it lacks clarity about research gap, novelty, a weak theoretical foundation, and the methodology employed does not sufficiently address the research questions, limiting its potential impact on the academic community. After incorporating all the changes and/or providing rebuttal statements where you have strong justification, kindly resubmit the manuscript for further processing.

Reviewers' comments:

Reviewer's Responses to Questions

**Comments to the Author**

1. Is the manuscript technically sound, and do the data support the conclusions?

Reviewer #1: Yes

Reviewer #2: Partly

2. Has the statistical analysis been performed appropriately and rigorously? 

Reviewer #1: Yes

Reviewer #2: No

3. Have the authors made all data underlying the findings in their manuscript fully available?

Reviewer #1: Yes

Reviewer #2: No

4. Is the manuscript presented in an intelligible fashion and written in standard English?

Reviewer #1: Yes

Reviewer #2: Yes

5. Review Comments to the Author

Reviewer #1: An interesting and very significant article to evaluates dynamic changes and spatial impact of production factors in China’s agro-processing industry agglomeration.

There are some minor structural comments that I recommend the authors to please integrate.

1. Line 97: "Agglomeration ,quantitatively" Please check comma and spacing

2. Line: 98: "Heterogeneity.This" Please correct spacing

3. Line 108: "labor.With the popularization" Please correct spacing

4. Line 185: "3. Model Construction and Indicator Selection 3.1 Empirical Model Construction". Please add brief introduction to main section before moving to subsections

5. Line 250: Numbering format, please correct numbering format

6. Line 336: "4. Empirical Results and Analysis 4.1 Current Status of Agglomeration" Please add brief introduction to main section before moving to subsections

7. Line 560: "5. Main Conclusions and Policy Relevance 5.1 Main Conclusion" Rephrase heading as "Conclusion" instead of "Main conclusions".

8. Line 590: Instead of adding separate title and numbers in conclusion section, you can add a sentence which can be linked to policy relevance.

Reviewer #2: Dear Bin Liu,

Thank you for submitting your research article titled "Research on the Dynamic Changes and Spatial Impact of Production Factors in China’s Agro-Processing Industry Agglomeration" to PLOS Journal. We appreciate the effort you have put into your study; however, after a thorough evaluation, we regret to inform you that your manuscript does not meet the criteria and domain of our journal.

Specifically, we have identified several key reasons for our decision to reject the manuscript:

Relevance to PLOS Journal: While your research focuses on the dynamic changes and spatial impact of production factors in China's agro-processing industry agglomeration, we find that the scope and subject matter are not in alignment with the areas of interest that PLOS Journal covers. Our journal primarily publishes research in the fields of biological and medical sciences, as well as some interdisciplinary studies. As such, we believe your manuscript would be better suited for a journal that specializes in economic or agro-industrial research.

Originality and Contribution: While it is evident that considerable effort has been invested in your study, we found the level of novelty and original contribution to be insufficient for publication in PLOS Journal. The research lacks a strong theoretical foundation, and the methodology employed does not sufficiently address the research questions, limiting its potential impact on the academic community.

Clarity and Structure: The presentation of your manuscript requires significant improvement in terms of clarity, organization, and language usage. The writing style and structure should adhere to the rigorous standards expected of scholarly articles, which are essential for facilitating clear communication and comprehension by our readers.

Abstract: Abstract is ambiguous. Abstract is not formulated in good way. A reader unable to understand how author is connecting agglomeration and dynamic changes in production factors against each other. Author did not mention how this research is unique from previous ones. Authors unable to give clarity among relationships between agglomeration and production factors. There is no need to mention (1), (2) in abstract.

Figures: Empirical results and analysis showed in form of figures are not presented in coherent way. Research article lacks essential bar graphs that could have provided a visual representation of the data and facilitated a clearer understanding of the relationships between agglomeration and different production factors. Graphical representations are crucial in conveying complex information effectively, and their absence hinders the comprehensibility and impact of your findings.

The graphs that were included in the manuscript did not sufficiently compare agglomeration and different production factors. Moreover, the absence of clear parameters or quotients on which these graphs were constructed raises concerns about the validity and reliability of the presented data. Transparently depicting the basis for your graphs is crucial for ensuring the credibility of your research.

Based on these grounds, we regretfully cannot proceed with the publication of your manuscript in PLOS Journal. However, we encourage you to revise your research for submission to a journal more closely aligned with your study's focus. Please consider thoroughly addressing the issues raised, improving the manuscript's structure, and clearly demonstrating the significance of your findings to enhance its chances of acceptance elsewhere.

6. PLOS authors have the option to publish the peer review history of their article (what does this mean?). If published, this will include your full peer review and any attached files.

Reviewer #1: No

Reviewer #2: **Yes: **Komal Imran

---

## [Author Response · Author response to Decision Letter 0]

11 Sep 2023

Responses to Reviewers' comments:

PLOS ONE Journal

Title: Research on the Dynamic Changes of China’s Agro-Processing Industry Agglomeration and Spatial Impact of Production Factors on Agglomeration

Manuscript Number: PONE-D-23-19856

Thanks for respected Reviewers’ useful comments and suggestions on this manuscript. We have modified this manuscript accordingly, and carefully. The revised manuscript has been uploaded as an attachment. Please view the modified content of the article in the revision mode in Word.

Here is the list corresponding to corrections point by point:

Reviewer# 1

1.Line 97: "Agglomeration, quantitatively" Please check comma and spacing

Our response: Thanks very much for the Reviewer’s kindly reminder. We have carefully revised the comma and maintained a reasonable spacing between sentences. The revised portions are marked in Main text, Line 95, Page 3 of this manuscript. 

2.Line: 98: “Heterogeneity. This” Please correct spacing

Our response: Thanks very much for the Reviewer’s kindly reminder. We replaced this section with a literature review and also maintained a reasonable spacing between sentences throughout the article. The revised portions are marked in Main text, Line 77-88, Page 2-3 of this manuscript.

3. Line 108: "labor. With the popularization" Please correct spacing

Our response: Thanks very much for the Reviewer’s kindly reminder. We have maintained a reasonable spacing between sentences. The revised portions are marked in Main text, Line 124, Page 3 of this manuscript. 

4. Line 185: "3. Model Construction and Indicator Selection 3.1 Empirical Model Construction". Please add brief introduction to main section before moving to subsections

Our response: Thanks very much for the Reviewer’s good suggestions. We have added a brief introduction between the main section and subsections, which made this part more coherent. The revised portions are marked in Main text, Line 199-203, Page 5 of this manuscript.

5. Line 250: Numbering format, please correct numbering format

Our response: Thanks very much for the Reviewer’s kindly reminder. The group headings were modified according to the template board format of the journal and numbering format of the article were standardized. The revised portions are marked of the manuscript.

6. Line 336: "4. Empirical Results and Analysis 4.1 Current Status of Agglomeration" Please add brief introduction to main section before moving to subsections

Our response: Thanks very much for the Reviewer’s good suggestions. We have added a brief introduction between the main section and subsections, which made this part more coherent. The revised portions are marked in Main text, Line 341-344, Page 9 of the manuscript.

7. Line 560: "5. Main Conclusions and Policy Relevance 5.1 Main Conclusion" Rephrase heading as "Conclusion" instead of "Main conclusions".

Our response: Thanks very much for the Reviewer’s good comments. We have corrected the rephrase heading as “Conclusion”. The revised portions are marked in Main text, Line 549, Page 16 of the manuscript.

8. Line 590: Instead of adding separate title and numbers in conclusion section, you can add a sentence which can be linked to policy relevance.

Our response: Thanks very much for the Reviewer’s kindly reminder. We have added a sentence between the conclusion section and policy relevance section, which make these two parts more coherent. The revised portions are marked in Main text, Line 573-575, Page 16 of the manuscript.

Reviewer# 2

1.Relevance to PLOS Journal

Our response: Thanks very much for the Reviewer’s kindly reminder.

（1）The agro-processing industry is an important part of the national economic system, which comprises more than 10 subsectors and accounts for a certain proportion of the manufacturing industry. Furthermore, the agro- processing industry is the key and main source of power to promote the industrialization of agriculture, and also is an important way to extend the agricultural industry chain, enhance the agricultural value chain and improve the agricultural supply chain. 

（2）Industrial agglomeration, as a common form of industrial organization, can economically bring about scale effects and achieve a more specialized division of labor and a more flexible and efficient allocation of resources, thus greatly reducing costs. China has promoted the accelerated development of the agro-processing industry in the main producing areas, forming a number of advantageous industrial clusters.

（3） Incremental inputs of capital, labor, technology and other relevant means of production are important sources of growth in the agricultural economy. While excessive investment in production factors may cause crowding effect. 

Based on the above reasons, it is of some practical significance for this article to study on dynamic changes of industrial agglomeration as well as explore the correlation between production factors and agglomeration of agro-processing industries. We believe that this article fits the research scope and subject of PLOS Journal to a certain extent and will attract more readers. 

In addition, PLOS Journal has published several articles related to industrial agglomeration, such as:

[1] Zhang QQ, Shi FJ, Abdullahi NM, Shao LQ, Huo X. An empirical study on spatial-temporal dynamics and influencing factors of apple production in China. PLOS ONE. 2020; 15(10). DOI10.1371/journal.pone.0240140

Main content: This article revealed the characteristics of spatial-temporal dynamics and influencing factors of China’s apple production layout by production concentration index and spatial econometric model.

[2] Han CJ, Wang GG, Zhang YX, Song LL, Zhu LZ. Analysis of the temporal and spatial evolution characteristics and influencing factors of China's herbivorous animal husbandry industry. PLOS ONE. 2020; 15(8). DOI10.1371/journal.pone.0237827

Main content: This article explored the spatial and temporal evolution characteristics and influencing factors of herbivorous animal husbandry industry based on the spatial autocorrelation analysis, standard deviation ellipse, and spatial Durbin model.

[3] Liao ZJ, Zhang LJ. Spatial distribution evolution and accessibility of A-level scenic spots in Guangdong Province from the perspective of quantitative geography. PLOS ONE. 2022; 16(11). DOI10.1371/journal.pone.0257400

Main content: The article used quantitative geography and geographic information system spatial analysis methods and analyzed the evolution of spatial distribution and regional accessibility of A-level scenic spots in Guangdong Province from 2001 to 2020. 

[4] Shi ZY, Xu DH, Xu LD. Spatiotemporal characteristics and impact mechanism of high-quality development of cultural tourism in the Yangtze River Delta urban agglomeration. PLOS ONE.2021; 16(6). DOI10.1371/journal.pone.0252842

Main content: The article studied the spatiotemporal characteristics and impact mechanism of the HDCT in the Yangtze River Delta by using coupling coordination degree model, Lisa spatiotemporal transition and spatial Durbin model (SDM).

[5] Li G, Yang Y, Lou XM, Wei YJ, Huang SF. Evaluation and spatial agglomeration analysis of the green competitiveness of China's manufacturing industry at the provincial level. PLOS ONE. 2021;16(3). DOI10.1371/journal.pone.0246351

Main content: This study uses Moran's I index to investigate the spatial agglomeration effect of the green development of the manufacturing industry at the province level.

2.Originality and Contribution 

Our response: Thanks very much for the Reviewer’s good comments. Compared with the previous literature, the marginal contribution of this study can be found in the following aspects:

(1)According to the judging criterion “Whether agricultural resources are directly used as production inputs?”, we redefined the scope of agro-processing industry by excluding industries with low agricultural relevance, such as printing and record media duplication, while including the traditional Chinese medicine manufacturing industry in the indicator research. At last, we obtained statistics for 11 sub-industries in 29 provinces of China. This is a new entry point for this topic.

(2)We analyzed the dynamics changes of China’s agro-processing industry agglomeration from 2001 to 2020, which clearly presented the layout and distribution of agro-processing industry in the past two decades. Meanwhile, we have distinguished differences in industrial agglomeration across sub-industries as well as across regions. Compared to previous literature, the study spanned a larger period of time and covered more sub-industries of the agro-processing industry.

(3)Unlike the previous literature that studied the factors influencing industrial agglomeration, we focus on the spatial impact of production factors on industrial agglomeration by revealing the direction of action and degree of effect basing on the dynamic spatial Durbin model, which provides reasonable suggestions and theoretical basis for the relevant departments responsible for the allocation of resources to the agro-processing industry, and promotes the high-quality development of agriculture. This is another new attempt in this article.

3.Clarity and Structure

Our response: Thanks very much for the Reviewer’s good comments. 

(1)To make the structure of this article clearer and help readers better 

understand, we have made improvements in the following areas: First, 

we have corrected the article title as “Research on the Dynamic Changes of China’s Agro-Processing Industry Agglomeration and Spatial Impact of Production Factors on Agglomeration” to better articulate the main body of the text. This article consists of two main parts, one is about the dynamic changes of the agglomeration of China’s agro-processing industry, the other is about exploring the spatial impacts of capital, labor, and technological production factors on the agglomeration of agro-processing industry in the long and short term, respectively. The revised portions are marked in Title, Line 1-2, Page 1 of the manuscript.

Second, we have provided a more detailed description of the research context, focusing on the theoretical and practical implications of this study. We have combed and reviewed the relevant national and international literature so as to find the entry point of the study in this article. The revised portions are marked in Main text, Line 57-88, Page 2-3 of the manuscript. 

(2)In order to make the theoretical foundation of this paper more solid, 

we have cited domestic and foreign classic theories and literatures to justify the theoretical mechanisms of this article. The revised portions are marked in Main text, Line 104-128, Page 3-5 of the manuscript.

(3)We constructed a dynamic spatial Durbin model to study the spatial 

correlation between production factors and agro-processing industry agglomeration, which is a widely used spatial econometric methods to study on industrial agglomeration. We could provide some literatures with high citations which used the spatial Durbin model to explain the correlation between industrial agglomeration and other factors as following: 

[1] Liu X, Zhang X. Industrial Agglomeration, Technological Innovation

and Carbon Productivity: Evidence from China. Resources Conservation and Recycling.2021;166. https://doi.org/10.1016/j.resconrec.2020.105330

[2] Yuan HX, Feng YD, Lee CC, Cen Y. How does manufacturing agglomeration affect green economic efficiency? Energy Economics.2022; 92. DOI10.1016/j.eneco.2020.104944

[3] Chen CF, Sun YW, Lan QX, Jiang F. Impacts of industrial agglomeration on pollution and ecological efficiency-A spatial econometric analysis based on a big panel dataset of China's 259 cities. Journal of Cleaner Production. 2020; 258. DOI10.1016/j.jclepro.2020.120721.

[4] Xu BK, Sun YM. The Impact of Industrial Agglomeration on Urban Land Green Use Efficiency and Its Spatio-Temporal Pattern: Evidence from 283 Cities in China. Land. 2023; 12(4). DOI10.3390/land12040824

(4) We have entrusted the article to a specialized editorializing agency to 

further refine the language. We also invited several experts to provide some advices, hoping to improve manuscript in terms of language usage. The revised portions are marked in the manuscript.

4.Abstract

Our response: Thanks very much for the Reviewer’s good suggestions. We have revised the abstract section by reorganized the main points of the article and provided full empirical results, which can better articulate the research in this article. The revised portions are marked in Abstract, Line 14-30, Page 1 of the manuscript.

5.Figures

Our response: Thanks very much for the Reviewer’s good suggestions. We have added two bar graphs instead of line charts (Fig1 and Fig2), which provided visual representation of China's agro-processing industry agglomeration from different subdivided industries and regions. The revised portions are marked in Main text, Page 8-9 of the manuscript.

6.Graphs

Our response: Thanks very much for the Reviewer’s good comments. 

This article contains 10 tables. We will introduce the specifics of these 10 Tables in detail.

(1)Before adopting the dynamic Durbin model with double fixed 

effects, the data need to be tested for LM test, LR test, Hausman test etc. to validate its scientific validity using this measure. Therefore, Table 1 was used to present results of Moran’s index, LM test, and robustness statistics. Table 2 was used to present results of LR, Hausman, and significance test. Table 3 was used to present results test results of bidirectional fixed effect. 

(2)Tables 4- Tables 6 contained the main results of this study by 

describing the direction and spatial effects of production factors and industrial agglomeration. Table 4 was used to demonstrate the relationship between production factor inputs and industrial agglomeration. Table 5 was used to show spatial spillover effect of production factors on industrial agglomeration in the short term. Table 6 was used to show spatial spillover effect of production factors on industrial agglomeration in the long term. 

(3)In order to avoid endogeneity of the measurements and to improve 

the reliability of the findings, the data need to be further validated by heterogeneity checking and robustness checking. Table A1- Table A4 in Appendix were to used to show the results of the heterogeneity test and the robustness test. Table A1 and Table A2 was used to reveal the different impact of production factor input on agglomeration from the perspectives of primary and deep agricultural processing industries, respectively. Table A3 was used to present influence of production factor input on industrial agglomeration from 2001 to 2015. Table A4 was used to show influence of production factor inputs on industrial agglomeration by replacing the dependent variable.

In addtion, to guarantee the credibility of the study, we have uploaded raw data suitable for Stata software as a separate file labeled 'agro-processing industry' and a documentation labeled ‘running code’.

Special thanks to you for your good comments and suggestions.

We tried our best to improve the manuscript and made some changes in the revised manuscript. We appreciate for Editors and Reviewers’ warm work earnestly, and hope that the correction will meet your approval.

Once again, thanks very much for your good comments and suggestions.

---

## [Decision Letter · Decision Letter 1]

2 Oct 2023

Research on the Dynamic Changes of China’s Agro-Processing Industry Agglomeration and Spatial Impact of Production Factors on Agglomeration

PONE-D-23-19856R1

Dear Dr. Liu,

We’re pleased to inform you that your manuscript has been judged scientifically suitable for publication and will be formally accepted for publication once it meets all outstanding technical requirements.

Kind regards,

Shujahat Haider Hashmi, PhD Regional Economics

Academic Editor

PLOS ONE

Additional Editor Comments (optional):

The reviewers have now thoroughly reviewed your revisions made in the paper and they recommend acceptance of your work. We appreciate your efforts and due patience in going through the stringent review process of our journal. We wish you best of luck for your future research endeavors.

Reviewers' comments:

Reviewer's Responses to Questions

**Comments to the Author**

1. If the authors have adequately addressed your comments raised in a previous round of review and you feel that this manuscript is now acceptable for publication, you may indicate that here to bypass the “Comments to the Author” section, enter your conflict of interest statement in the “Confidential to Editor” section, and submit your "Accept" recommendation.

Reviewer #1: All comments have been addressed

Reviewer #2: All comments have been addressed

2. Is the manuscript technically sound, and do the data support the conclusions?

Reviewer #1: Yes

Reviewer #2: Yes

3. Has the statistical analysis been performed appropriately and rigorously? 

Reviewer #1: Yes

Reviewer #2: Yes

4. Have the authors made all data underlying the findings in their manuscript fully available?

Reviewer #1: Yes

Reviewer #2: Yes

5. Is the manuscript presented in an intelligible fashion and written in standard English?

Reviewer #1: Yes

Reviewer #2: Yes

6. Review Comments to the Author

Reviewer #1: Thank you for addressing all the concerns. In the light of the changes made to the manuscript, I recommend manuscript for publication.

Reviewer #2: The authors has successfully address all the comments reasonably and updated review according to suggestion. I am satisfied with all the sections and agreed for acceptance of article in PLOS ONE.

7. PLOS authors have the option to publish the peer review history of their article (what does this mean?). If published, this will include your full peer review and any attached files.

Reviewer #1: No

Reviewer #2: **Yes: **Komal Imran

---

## [Editor Report · Acceptance letter]

13 Oct 2023

PONE-D-23-19856R1 

Research on the dynamic changes of China’s agro-processing industry agglomeration and spatial impact of production factors on agglomeration 

Dear Dr. Liu:

I'm pleased to inform you that your manuscript has been deemed suitable for publication in PLOS ONE. Congratulations! Your manuscript is now with our production department. 

Kind regards, 

on behalf of

Dr. Shujahat Haider Hashmi 

Academic Editor

PLOS ONE